# Anti-Melanogenesis Effect of Polysaccharide from *Saussurea involucrata* on Forskolin-Induced Melanogenesis in B16F10 Melanoma Cells

**DOI:** 10.3390/nu14235044

**Published:** 2022-11-27

**Authors:** Mayila Kamilijiang, Deng Zang, Nuermaimaiti Abudukelimu, Nurbolat Aidarhan, Geyu Liu, Haji Akber Aisa

**Affiliations:** 1State Key Laboratory Basis of Xinjiang Indigenous Medicinal Plants Resource Utilization, Xinjiang Technical Institute of Physics and Chemistry, Chinese Academy of Sciences, Urumqi 830011, China; 2University of Chinese Academy of Sciences, Beijing 100049, China

**Keywords:** anti-melanogenesis, B16F10 melanoma cell, *Saussurea involucrata*, β-catenin degradation

## Abstract

As one of the prominent medicinal plants listed in the Chinese pharmacopoeia (2020), *Saussurea involucrata* (Kar. et Kir.) Sch.-Bip was demonstrated to possess various therapeutic effects. In our recent research, we extracted the polysaccharides from *S. involucrata* (SIP) at optimal conditions and conducted further structure elucidation on the main fraction as well as the confirmation of its possible anti-inflammatory activity. Hence, in this work, we assessed the in vitro antioxidant activity and anti-melanogenesis effects of the crude SIP in forskolin-induced B16F10 melanoma cells. The results show that SIP possessed strong antioxidant activity and was effective in concentration-dependently decreasing melanin formation and inhibiting tyrosinase activity in forskolin-induced B16F10 cells. Based on these results, the inhibitory mechanism of melanogenesis was investigated by measuring Tyrosinase (TYR), Tyrosinase related protein-1 (TRP-1), Tyrosinase related protein-2 (TRP-2), Microphthalmia-associated transcription factor (MITF), cAMP-response element binding protein (CREB), mitogen-activated protein kinases (MAPK) signaling protein members, and β-catenin degradation in forskolin-induced B16F10 cells. The anti-melanogenesis response of SIP might be attributed to the regulation of c-Jun *N*-terminal kinase (JNK) phosphorylation and β-catenin degradation pathways. These results suggest that polysaccharides from *S. involucrata* possess a strong anti-melanogenic effect, and thus could be used as a high-value natural material for skin whitening in cosmeceutical industries.

## 1. Introduction

Melanocytes are known as the specialized skin cells mainly found in the epidermis. Melanocytes produce specific organelles, named melanosomes, where melanin is produced and deposited. When melanocytes are triggered by external elements or other biological factors, melanin is synthesized in the melanosome and transferred from the dendritic terminals of the melanocytes to the nearby keratinocytes, a process known as melanogenesis [1]. The pigment melanin is responsible for determining a person’s skin, eye, and hair color. In addition, this pigment shields the skin cells from the detrimental influence of solar ultraviolet rays, oxidative stress, and other contaminants found in the environment [2,3,4]. However, excessive synthesis of melanin can cause a variety of skin diseases, such as freckles, age spots, melasma, dark skin disease, and other hyperpigmentation syndromes [5,6,7]. Melanin synthesis is a highly complex process and numerous chemical and enzymatic events occur inside the melanosomes throughout the process of melanogenesis. Generally, in melanocytes, tyrosinase (TYR), a copper-containing enzyme catalyzes the substrate tyrosine to the intermediate product L-3,4-dihydroxyphenylalanine (L-DOPA). After that, L-DOPA is converted into two different melanin pigments, eumelanin and pheomelanin, through other interrelated enzymatic reactions. In this process, TYR performs as a rate-limiting enzyme, catalyzing the production of melanin and other pigments from tyrosine by oxidation. Tyrosinase-related protein 1 (TRP-1) and tyrosinase-related protein 2 (TRP-2) are melanocyte-specific gene products also involved in the production of melanin formation via participating in a series of enzymatic reactions [8,9,10,11]. Melanocytes regulate melanin synthesis by autocrine and paracrine factors, as well as numerous external stimuli such as ultraviolet radiation. Recent research suggests that keratinocytes control the production of melanin by secreting various cytokines, including endothelin-1 and α-melanocyte-stimulating hormone (α-MSH). Additionally, cells such as fibroblasts and immune cells can actively regulate melanin formation by secreting paracrine chemicals [12,13]. These elements regulate the formation of melanin by affecting the microphthalmia-associated transcription factor (MITF), a master element that regulates pigmentation [12]. Research suggests that by binding to the M-box in the TYR family promoter region, MITF functions as a transcription factor to control the transcription of TYR, TRP-1, and TRP-2 [14,15].

Numerous signaling molecules can control the transcription of the MITF gene or regulate it at the post-translational level. The cAMP/PKA signaling pathway is regarded as the classical signaling pathway that regulates melanocyte proliferation and participates in the melanin synthesis process [16]. Additionally, mitogen-activated protein kinases (MAPKs) consisting of the extracellular signal-regulated kinase (ERK) [17,18,19], c-Jun *N*-terminal kinase (JNK) [20], p38 MAPKs [21], and phosphatidylinositol 3 kinase (PI3K)/Akt and Wnt signaling pathways are all playing significant roles in melanogenesis through modulating MITF [16]. In addition, proteasomal mechanisms that are involved in the elimination of aberrant or inaccurate-structured proteins throughout the ubiquitination and degradation process can also regulate melanogenic proteins [22,23]. MITF, TYR, or β-catenin were previously found to be eliminated by the ubiquitin-mediated proteasomal degradation pathway [13,24,25,26,27,28]. Therefore, some skin-whitening substances significantly contribute to the regulation of pigmentation by the degradation of MITF transcription factors and/or melanogenic proteins.

Hydroquinone, ascorbic acid, kojic acid, and arbutin are a few hypopigmented substances that were identified as melanin inhibitors. However, these described traditional pigment inhibitors have the disadvantage of obvious cytotoxicity, strong irritation, high allergenicity, and significant unpleasant responses, making it challenging to meet the needs of the majority of consumers [29,30]. Recent cosmetic research has therefore paid attention to creating innovative skin-lighting substances that are safe and efficient melanin inhibitors derived from natural products and traditional Chinese medicine [5,31,32].

The *Saussurea* is classified under the *Compositae* family consisting of around 400 species, most of which are grown in chilly climates around the world. China has the most diversity of 289 species [33]. It is a rare herb that grows at elevations between 2400 and 4100 m in meadows, rock fissures, and mountain slopes and was designated as a second-class national preserved wild plant in China [34]. The dried ground part of *Saussurea involucrata* (Kar. et Kir.) Sch.-Bip. is known as *S. involucrata* and “Snow lotus” (Xue lian in Chinese), mainly distributed in the Tianshan and Kunlun Mountains in Xinjiang. The entire herb, which contains a range of volatile oils, alkaloids, flavonoids, phenols, tannins, and other compounds, can be utilized as medication [35]. *S. involucrata* was used as a crucial medicinal herb by many ethnomedical systems, including traditional Chinese medicine, Uyghur medicine, Kazakhstan medicine, and Mongolian medicine, to disperse cold and remove dampness, activate blood circulation, act as anti-inflammatory and analgesic ingredients, and treat rheumatic arthritis [34]. The bioactive components found in *S. involucrata*, including flavones, phenylpropanoids, polysaccharides, lignans, and sesquiterpenoids, were linked to some bioactivities, including antioxidant, anti-UV, and anti-aging [35,36,37,38,39,40]. In a previous study, we isolated the polysaccharide from *S. involucrata* and found that it possessed great anti-inflammatory activity, indicating its potential use in the functional food, pharmaceutical, and cosmetical industry [41].

Studies have demonstrated that polysaccharides can be used as skin-brightening agents because of their antioxidant and anti-melanogenesis properties [42,43,44]. Lei et al. separated the sulfated polysaccharides from the Celluclast-assisted extract of *Hizikia fusiforme*, and verified its anti-melanogenesis effects by blocking the TYR family protein expressions in B16F10 melanoma cells [42]. Physicochemical properties of pectin from wampee fruit were found to be able to block α-MSH-stimulated melanogenesis in A375 melanoma cells by the α-MSH signaling pathway, according to a study by Weiyu [43]. Sulfated polysaccharides extracted from *Ecklonia maxima*, a kind of brown seaweed, were reported to significantly decrease melanin production and suppressed tyrosinase activity in α-MSH-induced B16F10 cells, suggesting their potential use in pharmaceutical and cosmeceutical applications [45]. However, few studies have investigated the skin pigment-regulating effect of the polysaccharides from *S. involucrata*. Therefore, to further exploit the application of SIP in the cosmetics industry, *in vitro* antioxidant, and anti-melanogenesis actions of SIP and its underlying mechanism were evaluated using forskolin-stimulated B16F10 melanoma cells in this study. To our knowledge, this may be the first time a comprehensive investigation of the anti-melanogenesis activity of polysaccharide from *S. involucrata* was conducted. These findings might provide the viability of employing SIP as a natural skin-whitening agent in cosmetics.

## 2. Materials and Methods

### 2.1. Chemicals

L-3,4-dihydroxyphenylalanine (L-DOPA), ABTS, DPPH, and mushroom TYR were purchased from Sigma-Aldrich (St. Louis, MO, USA). Forskolin and MG132 (Lot: 119850) were purchased from MCE (MedChemExpress, Monmouth Junction, NJ, USA). Kojic acid (Cat#K8070) was purchased from Solarbio (Beijing, China). Primary antibodies for TYR family proteins were obtained from Santa Cruz Biotechnology (Dallas, TX, USA). Furthermore, antibodies for MITF, phospho-GSK-3β (*p*-GSK), GSK-3β, β-catenin, *p*-JNK, JNK, *p*-P38, P38, *p*-CREB, *p*-ERK, ERK were purchased from Cell Signalling Technology (Beverly, MA, USA). Antibodies for Lamin B were obtained from Absin Bioscience Inc. (Shanghai, China). β-actin and antibodies designed to target primary antibodies were obtained from BOSTER Biological Technology (Wuhan, China). The Nuclear and Cytoplasmic extraction kit used for separating nuclear protein extract from the cytoplasmic fraction was obtained from Beyotime Technology (Shanghai, China).

### 2.2. Sample Preparation

*S. involucrata* polysaccharide (SIP) was prepared in our prior research and preserved at the State Key Laboratory Basis of Xinjiang Indigenous Medicinal Plants Resource Utilization. The methods for separation and analysis were described by Liu et al. [41]. Briefly, the *S. involucrata* was dried and defatted with petroleum ether (1:10, *w*/*v*) 3 times for 2 h followed by air-drying and de-coloring. Consequently, the polysaccharides were extracted at optimal conditions of 80 °C with a liquid-solid ratio of 30 mL/g and yield rate of 11.37% for 2 h.

### 2.3. In Vitro Antioxidant Activity

#### 2.3.1. DPPH Radical Scavenging Assay

We performed a DPPH free radical scavenging assay referring to the protocol reported by Li et al. [46] with a few minor adjustments. In brief, a DPPH stock solution of 2 mmol/L was prepared and kept in the dark at −20 °C. The stock solution was then diluted for the experiment until the working concentration absorbance was set at about 0.70 ± 0.02 at 515 nm. Next, 100 μL of sample solution was then mixed with 100 μL of DPPH solution in each well of a 96-well microplate. The absorbance was observed at 515 nm following 30 min of incubation at room temperature. Each concentration of the sample solution was tested in triplicate and ascorbic acid/Vitamin C was used as a positive control. A formula was applied to obtain the inhibition percentage: Inhibition percentage (%) = [1 − (OD samples − OD control)/OD blank] × 100%.

#### 2.3.2. ABTS Cation Radical Scavenging Assay 

With certain changes, the ABTS free radical scavenging test was performed following the method by Bae [47] with certain changes. An ABTS assay working reagent was made of potassium persulfate and ABTS stock solution with a certain ratio. The working solution was stored at an indoor temperature and avoided light for 16 to 18 h. Absolute ethanol was applied to dilute the ABTS solution, to set the working concentration absorbance to 0.70 ± 0.02 at 734 nm. The assay was carried out in a 96-well microplate by mixing 16 μL of various concentrations of sample solution with 184 μL of ABTS working solution immediately and thereafter incubated for nearly 5 min at room temperature until the solution color changed, and the absorbance of all tested wells was measured at 734 nm. The desired concentration of the sample solution was measured in triplicate and ascorbic acid/Vitamin C was utilized as a positive control. Using the following formula, the inhibition % was determined: Inhibition percentage (%) = [1 − (OD samples − OD control)/OD blank] × 100%.

### 2.4. Anti-Melanogenesis Activity Evaluation

#### 2.4.1. Tyrosinase Inhibitory Assay

The mushroom tyrosinase inhibition assay was conducted using L-DOPA as a reaction substrate as stated in reference [48] with some modifications. Briefly, 40 μL of PBS (10 mM, pH 6.0) was added into each well of the 96-well plate as the system buffer. The required quantities of sample solution were added followed by 25 U/mL of mushroom tyrosinase for each well. After that, each well received L-DOPA (10 mM) and was incubated at 37 °C for 10 to 20 min. A microplate reader, Spectra Max M5 (Molecular Devices Company, San Jose, CA, USA) was used to measure the relative amount of DOPA quinone produced at an absorbance of 490 nm. As the positive control, kojic acid (KA), a well-known inhibitor of tyrosinase, was employed. The formula used to determine the tyrosinase inhibition is as follows: Tyrosinase inhibition (%) = (1 − (OD samples − OD blank)/(OD control − OD blank)) × 100.

#### 2.4.2. Cell Culture and Cell Viability Assay

The Kunming Cell Bank of Type Culture Collection of the Chinese Academy of Sciences (Kunming, China) is where the B16F10 murine melanoma cells used in this investigation were purchased. The Dulbecco’s modified Eagle medium (Gibco, Grand Island, NY, USA) with 10% fetal bovine serum (Biological Industries, Kibbutz Beit-Haemek, Israel) and 1% penicillin/streptomycin was used as the culture media for the B16F10 melanoma cells. The MTT assay was used to evaluate the cytotoxic activity of the tested sample. Briefly, B16F10 cells were seeded in 96-well plates (1 × 10^4^ cells/well) for overnight incubation. Thereafter, the medium was changed to a medium that included SIP in a range of concentrations (0 as the control, 0.05, 0.1, 0.5, 1, 2, or 4 mg/mL). After the desired time of incubation, 100 μL of concentrated MTT solution (0.5 mg/mL) was added to each well in the media. The plates were covered in aluminum foil and kept in the incubator for 4 h at 37 °C. The formazan was finally dissolved using DMSO after the medium had been removed. The absorbance of the formazan concentration was read at 490 nm.

#### 2.4.3. Measurement of Cellular Melanin Contents

The melanin content of B16F10 cells was measured by referring to the protocol described by Zang et al. [49]. Briefly, B16F10 melanoma cells were seeded at a density of 0.7 × 10^5^ cells/well onto a 6-well plate and incubated until the following day. The medium was changed for a new fresh medium containing positive control kojic acid (700 μM) and SIP (0.1, 0.5, 1 mg/mL) in the presence of forskolin (4 μM) and incubated for another 72 h. Next, the media was then taken out, and PBS was used to rinse the cells. Then, the cells were harvested with RIPA. The harvested cells were centrifuged at 12,000 rpm, 4 °C for 22 min. The supernatants were removed leaving the cell pellets as dry as possible. The total protein in the supernatants of cell extracts was measured by the BCA protein assay kit. NaOH (200 μL, 1 M) containing 10% DMSO was used to solubilize the remaining cell pellets at 80 °C for 60 min. Next, the dissolved cell lysates were added to a 96-well plate and the characteristic absorption peak was tested at 405 nm. Melanin formation was quantified by adjusting the overall amount of melanin content to the corresponding protein concentration. Each experiment was performed three times.

#### 2.4.4. Intracellular Tyrosinase Activity Assay

The method for intracellular TYR activity assay mentioned in the research by Zang et al. [49] was slightly modified in our experiment. Briefly, a 6-well plate was seeded with B16F10 melanoma cells at a density of 1 × 10^5^/well. The following day, the medium was changed for a new fresh medium containing positive control kojic acid (700 μM) and SIP (0.1, 0.5, 1 mg/mL) in the presence of forskolin (4 μM) and incubated for another 24 h. Next, sodium deoxycholate was dissolved in PBS containing 1%TritonX-100 to harvest the rinsed cells. The harvested cells were subsequently centrifuged for 22 min under the conditions of 12,000 rpm and 4 °C. After measuring the protein contents in supernatants of cell extracts, the residual supernatant was used to examine the intracellular tyrosinase activity in the manner described below: 96-well plates containing 60 μL of supernatant and 10 μL of L-DOPA were incubated at 37 °C for 10 to 20 min. Using a microplate reader, we determined the characteristic absorption at 490 nm.

### 2.5. Western Blot Analysis

The B16F10 melanoma cells were washed in PBS before being harvested on ice with RIPA buffer containing protease inhibitors. Proteins for Western blotting were then quantified using a BCA assay kit and denatured with loading buffer at 100 °C for 5 min. SDS-PAGE was used to separate the previously prepared proteins, and the proteins were subsequently transferred to PVDF membranes for 120 min at 400 mA. Commercial non-fat milk dissolved in 1×TBST was used to block the membrane, followed by incubating the membrane in primary antibody solution overnight at 4 °C with gentle rocking. The membrane was then rinsed and incubated for an additional hour at room temperature with the secondary antibodies. Finally, the membrane was photographed using the ChemiDocMP Imaging System (Bio-Rad Laboratories, Hercules, CA, USA) after rinsing and identified using chemiluminescence Western blotting detection reagents. Each experiment was run three times. Nucleus and cytoplasm protein separation experiments were performed using the Nucleoplasmic and Cytoplasmic Protein Extraction Kit following the manufacturer’s protocol.

### 2.6. Statistical Analysis

Results were presented as mean ± standard deviation (SD). One-way ANOVA and Tukey’s multiple comparison test were used for the statistical analysis by applying GraphPad Prism 9 (Dotmatics, Boston, MA, USA). The differences were considered statistically significant as *p*-values < 0.05.

## 3. Results

### 3.1. Radical Scavenging Activity

We evaluated the antioxidant effects of SIP using the ABTS/DPPH assay, with ascorbic acid as a positive control. The outcomes are displayed in Figure 1. The SIP’s ABTS radical scavenging activity at 1 mg/mL was 99.19%, which is comparable to the activity for positive control vitamin C (99.58%) under the same condition (Figure 1A). At a dosage of 0.087 mg/mL, SIP demonstrated a 50% suppression of the ABTS radical (IC50 value). In Figure 1B, at a concentration of 3 mg/mL, SIP displayed the highest DPPH radical scavenging activity at 90.69%, with an IC50 value of 0.112 mg/mL.

### 3.2. Mushroom Tyrosinase Inhibitory Assay

SIP was initially tested for its anti-melanogenesis ability utilizing a mushroom tyrosinase assay employing L-DOPA as a substrate in a cell-free environment. We discovered that SIP did not inhibit the activity of mushroom tyrosinase, as presented in Figure 2. However, the positive control used in the experiment, known as the tyrosinase inhibitor kojic acid, dramatically reduced the mushroom tyrosinase with an IC50 of 0.705 mM. Based on these findings, it was concluded that SIP has no direct impact on tyrosinase activity. Since numerous mechanisms can control melanin formation, we used SIP treatment at various dosages to examine its impact on melanogenesis in forskolin-stimulated B16F10 cells.

### 3.3. Effect of SIP on B16F10 Melanoma Cell Viability

The cytotoxicity of SIP was measured by the MTT assay prior to the measurement of melanin in B16F10 melanoma cells. The cells were intervened with SIP for 48 h at 0.05, 0.1, 0.5, 1, 2, and 4 mg/mL concentrations. As shown in Figure 3, after 48 h of incubation, SIP was not found to be cytotoxic to B16F10 melanoma cells at any of the desired concentrations. Therefore, further experiments were carried out by selecting 0.1, 0.5, and 1.0 mg/mL concentrations.

### 3.4. Effect of SIP on Melanin Content and Tyrosinase Activity

Subsequently, we used forskolin (FSK) to establish a melanin deposition cell model by stimulating melanin formation in B16F10 cells to examine the impact of SIP on melanin formation. Based on the cytotoxicity data, B16F10 cells were treated with various concentrations (0.1, 0.5, and 1 mg/mL) of SIP for 72 h in the presence of forskolin. Figure 4 depicts that the forskolin treatment increased melanin content to 238.55% compared to the control. Treatment with SIP caused a substantial decrease in the melanin content of B16F10 cells in a dose-dependent manner (Figure 4A). At a dose of 1 mg/mL, SIP showed the highest melanogenesis inhibitory effect (inhibition rate, 49.42%) compared to other tested SIP concentrations (0.1 or 0.5 mg/mL). Cells treated with SIP at 0.1 mg/mL exhibited approximately the same ability of melanin inhibition as cells treated with Kojic acid at 700 μM, a positive control. Furthermore, the color of cell lysates in the SIP-treated group at a concentration of 1 mg/mL was significantly lighter than the color of the only FSK-stimulated group (Figure 4A).

Additionally, we investigated how SIP affected intracellular tyrosinase activity in FSK-induced cells. According to the results in Figure 2B, the activity of tyrosinase in FSK exposure groups rose by about 23.2% compared to those that were not stimulated. Co-treatment with SIP potently inhibited such activity, and this effect was observed in a dose-dependent manner. Compared to the group that was only induced by FSK, SIP treatment at low, medium, and high concentrations suppressed tyrosine activity by 11%, 17%, and 19%, respectively. As a positive control, Kojic acid inhibited the tyrosinase activity by only 10% at 700 μM. Collectively, the patterns of SIP on melanin content and intracellular tyrosinase activity show similar results indicating that SIP can effectively suppress melanin synthesis in melanocytes.

### 3.5. Effect of SIP on Tyrosinase and Tyrosinase-Related Protein Expressions in Forskolin Stimulated B16F10 Cells

The main participants in the synthesis of melanin in the melanosome were proteins from the tyrosinase family: TYR, TRP-1, and TRP-2. To further elucidate the underlying mechanism, we investigated if the SIP-induced decrease in melanin production would be linked to the downregulation of those melanogenic regulatory proteins (Figure 5). The results revealed that forskolin treatment remarkably increased TYR expression, whereas SIP treatment significantly and dose-dependently inhibited TYR expression. Intriguingly, 1 mg/mL SIP treatment reduced TYR expression below the basal level compared to the control group. However, Kojic acid treatment did not block TYR under comparable stimulation conditions. TRP-1 was likewise discovered to have a comparable inhibitory impact on cells treated with SIP. Forskolin stimulation, however, did not successfully increase the expression of TYP-2 in B16F10 cells, while SIP administration still markedly reduced the expression of TRP-2 (Figure 5). Additionally, SIP show better potential than Kojic acid to suppress the expression of proteins involved in the production of melanin. Together, these results indicate that SIP-mediated melanin inhibition might be caused by the downregulation of TYR, TRP-1, and TRP-2.

### 3.6. Effects of SIP MITF and Phosphorylation of CREB Protein Expressions in Forskolin-Stimulated B16F10 Cells

It was proven that the TYR gene family proteins (TYR, TRP-1, and TRP-2) in melanocytes are transcriptionally modulated by MITF, which is known as the master regulator of pigment production. We subsequently examined MITF protein expression following SIP treatment. As depicted in Figure 6A, forskolin treatment resulted in a significant rise in MITF protein levels, whereas SIP resulted in a considerable decrease in MITF expression. SIP at a dosage of 0.1 mg/mL downregulated the MITF protein expression level by 85.88% compared with the forskolin-only stimulated group.

Forskolin, a direct cAMP activator, increases the cellular content of cAMP, which subsequently boosts the expression of the MITF via cAMP response element-binding protein (CREB) [50]. Therefore, in FSK-modulated B16F10 cells, we investigated SIP’s impact on the phosphorylation and total protein levels of CREB. Figure 6 depicts the effects of forskolin stimulation on CREB phosphorylation at Ser133 in B16F10 cells, whereas co-treatment with SIP at 0.1 mg/mL slightly down-regulated the levels of *p*-CREB by 46.4%. This result suggests that SIP partially inhibited MITF expression in B16F10 cells by preventing CREB-mediated MITF synthesis.

### 3.7. Effects of SIP on the Expression of MAPK Signaling Pathway in Forskolin-Stimulated B16F10 Cells

Studies show that the MAPK signaling pathway is also known to modulate the MITF expression, thus playing a significant role in melanin formation [40]. Therefore, the protein expression of central MAPK pathway members ERK, JNK, and p38 were examined at both total and phosphorylated levels in B16F10 cells in the presence or absence of SIP treatment to evaluate the potential function of MAPK in the control of melanogenesis. The expression of *p*-ERK and *p*-p38 proteins was not affected by forskolin or SIP treatment, as seen in Figure 7A. The protein expression level of *p*-JNK was 1.46 times higher in forskolin-stimulated melanoma cells than in the control group, whereas SIP treatment reduced the amount of *p*-JNK by 30.9% at doses of 1 mg/mL (Figure 7B). These findings demonstrate that SIP’s effects on melanin synthesis in melanocyte cells relate to JNK1/2 MAPK phosphorylation.

### 3.8. Effects of SIP on the β-Catenin Signaling Pathway

Considering that SIP suppressed MITF expression in B16F10 melanoma cells, we then investigated if β-catenin signaling is involved in this suppression. As the primary regulator of Wnt/β-catenin signaling, we first examined the SIP impact on intracellular β-catenin levels. Data show that the protein level of β-catenin was much higher in FSK-stimulated melanoma cells than in the non-treatment control group, while treatment with SIP remarkably as well as concentration-dependently decreased the intracellular β-catenin content in B16F10 melanoma cells.

According to earlier studies, proteasome-dependent proteolysis takes part in an essential role in regulating the amount of intracellular β-catenin [51,52]. We proceeded to determine if the down-regulation of β-catenin resulting from SIP was also caused by proteasome degradation. The β-catenin expression level generated by forskolin in B16F10 cells was consistently lowered by SIP, as shown in Figure 8A. However, the proteasome inhibitor MG132 completely reversed the impact of SIP on β-catenin decrease by blocking -catenin degradation (Figure 8B). Consistent with this, MG132 also increased the MITF protein level in B16F10 cells in the presence of SIP evidently. Together, our results indicated that SIP suppressed β-catenin signaling by boosting β-catenin degradation.

Additionally, we detected the β-catenin expression levels in the cytoplasm and nucleus of forskolin-induced B16F10 cells. Our findings suggest that SIP treatment decreased the level of β-catenin dramatically in the nucleus and cytoplasm (Figure 9).

## 4. Discussion

Melanogenesis is defined as an elaborate process that includes a variety of enzymatic and chemical reactions, which occurs in the specialized organelles known as melanosomes in melanocytes, resulting in the synthesis of cellular melanin. After being produced in the melanocytes, the melanin is transferred to the surrounding keratinocytes by the dendritic structure of the melanocytes, where it is reorganized and dispersed. The melanin granules travel upward with keratinocyte differentiation and eventually reach the stratum corneum, where melanin creates visible pigmentation [53,54]. Although melanin protects the skin by absorbing harmful rays and scavenging free radicals, excessive melanin production can cause a variety of significant dermatological conditions, including freckles, age spots, melasma, and other hyperpigmentation syndromes [29]. Most skin-whitening products used today, including hydroquinone, kojic acid, and hydrocortisone, are generally toxic compounds, and prolonged application can induce adverse effects, such as skin stimulus and itching [29,30]. Hence, it is essential to concentrate on the creation of natural skin-whitening products that are both safe and efficient and have few adverse effects.

*S. involucrata* is a dicotyledonous plant used as a traditional medicinal plant in ethnomedicine that was discovered to have several bioactivities including anti-cancer, anti-inflammation, anti-oxidation, adipogenesis suppression, and neuroprotection activities [34]. Our previous research showed that polysaccharides from *S. involucrata* possessed remarkable anti-inflammatory activity indicating their potential in the functional food, pharmaceutical, and cosmetical industries [41]. To our knowledge, the investigation of the cosmetic application of SIP is rare. Thus, in this work, we mainly concentrated on the anti-melanogenic property of SIP.

UV radiation can cause oxidative stress, which can induce melanin accumulation. Therefore, natural products with such free radical scavenging activity can inhibit melanin production thereby possessing potential whitening ability [48,55,56]. Therefore, in the first step, we evaluated the antioxidant activity of SIP using DPPH and ABTS assays, as they were widely used and straightforward approaches for assessing in vitro antioxidant activity. DPPH is a stabilized nitrogen-centered chromogenic radical which produces a characteristic absorption at 517 nm. Due to the presence of antioxidant substances in the sample, the absorbance of DPPH decreases, and the amount of change in absorbance is linearly related to the level of antioxidant substances within a specific period, thus can be used to quickly and precisely assess a plant extract’s antioxidant capacity [57,58]. When ABTS interacts with the appropriate oxidants, it forms blue-green ABTS cation radicals, which have a distinctive absorption peak at 734 nm. When antioxidative substances are present, the ABTS cation radical production is suppressed, which causes a decrease in absorbance. Both fat-soluble and water-soluble antioxidants can be used with this approach [46]. The results of the current DPPH and ABTS experiment demonstrate that SIP can scavenge free radicals, which is crucial when used as primary antioxidants in functional cosmetics.

Tyrosinase is a rate-limiting enzyme that is essential for the formation of melanin. Two processes are involved in tyrosinase-mediated melanin synthesis [59]. First, L-tyrosine or L-DOPA is hydroxylated; next, L-DOPA is changed into DOPA quinone. Tyrosinase activity must therefore be inhibited to decrease the generation of L-DOPA from L-tyrosine, which is a crucial step in inhibiting melanin formation. As a consequence, tyrosinase inhibitory action is an important factor to consider when evaluating skin-whitening products [60]. To initially investigate its anti-melanogenic activity, SIP was first screened for whitening using a mushroom tyrosinase inhibiting assay in a cell-free system with L-DOPA as the substrate, because it is efficient, convenient, and commercially available [14]. Our findings, however, indicated that SIP at the tested concentrations may not directly alter tyrosinase activity as there was no obvious inhibitory effect on mushroom tyrosinase activity.

As melanogenesis can also be regulated by various pathways, we further examined the effect of SIP on melanogenesis in forskolin-induced B16F10 cells. B16F10 melanoma cell lines are the preferred cell models for screening whitening compounds because of their advantage of being able to pass through numerous generations, rapid proliferation, relatively low culture condition needs, and sensitivity to chemical stimulation [16,61,62]. Initially, an MTT assay was applied to confirm the non-cytotoxic content range of SIP for B16F10 cells. After 48 h of incubation, the results showed that SIP was found not cytotoxic to B16F10 melanoma cells at any of the evaluated dosages (0.05–2.00 mg/mL). Therefore, 0.1, 0.5, and 1.0 mg/mL concentrations were selected for further experiments.

Several substances including α-MSH, 3-isobutyl-1-methylxanthine (IBMX), forskolin, and β-endorphin (β-END) can stimulate the melanogenesis of melanoma cells. Among them, forskolin is commonly used as the activator of adenylyl cyclase to increase the intracellular cAMP levels, thereby inducing melanogenesis [63,64]. In this study, we used forskolin to stimulate the formation of melanin and analyzed the melanin-inhibitory effects of SIP. We discovered that the impacts of SIP on melanin levels and intracellular tyrosinase activity had a similar pattern as both were dose-dependently reduced after SIP treatment. In particular, it was shown that melanin content dropped by over 50% at a dosage of 1 mg/mL of SIP compared to the group that had only been stimulated by FSK, demonstrating that SIP can effectively inhibit melanin formation in melanocytes (Figure 4A).

The tyrosinase family, specifically TYR, TRP1, and TRP2 are master regulators of the production of melanin and pigmentation. Therefore, to comprehend the possible mechanism for the observed decrease in melanin content brought on by SIP treatment, we first investigated how SIP affected the melanogenic enzyme protein expression. According to the Western blotting results, tyrosinase protein levels were shown to be decreased by SIP in a dose-dependent pattern. Particularly, at a dosage of 1 mg/mL, a substantial reduction in TYR protein expression was seen. A similar inhibitory effect of TRP-1 and TRP-2 was also found in SIP treatment. However, under comparable stimulation conditions, Kojic acid was unable to block TYR expression.

MITF is pivotal to the production and transport of melanin by regulating key melanogenic proteins, namely TYR, TRP-1, and TRP-2. As a transcription factor, it can interact with a highly conservative gene sequence called M-box shared by TYR, TRP-1, and TRP-2 in the promoter region [65]. This interaction affects TYR, TRP-1, and TRP-2 protein expressions via upregulating TYR family gene transcription [66]. In this work, we discovered that forskolin treatment significantly increased the protein level of MITF; however, SIP at a dose of 0.1 mg/mL downregulated the level of MITF expression by 85.88% compared to the forskolin-only stimulated group. These findings imply that, in B16F10 cells, SIP reduced tyrosinase and tyrosinase-related protein expression by regulating the expression level of MITF, which affects melanogenesis.

Except for regulating TYR, TRP-1, and TRP-2 protein expression levels, MITF itself can be affected by a variety of signaling molecules. The promoter region of MITF contains the binding sites for transcription factors such as PAX3 (paired box 3), SOX 10 (sex-determining region Y box 10), LEF1 (lymphoid enhancer binding factor 1), and CRE (cAMP response element) [20]. Forskolin, a direct cAMP activator, stimulates the synthesis of cAMP and further upregulates and activates the expression of CREB via phosphorylation. The phosphorylated CREB then connects with the promoter site of MITF and triggers the MITF expression [50]. Therefore, we investigated the effect of SIP on FSK-mediated CREB phosphorylation protein levels in B16F10 cells. Results revealed that SIP treatment slightly suppressed the FSK-caused phosphorylation of CREB. Taken together, it can be concluded that SIP suppressed MITF production in B16F10 cells partially by inhibiting CREB phosphorylation.

The MAPKs are a class of serine/threonine protein kinases that are commonly found in mammals and can be triggered by various extracellular signals or stimulants. MAPK family proteins such as p38, ERK, and JNK are crucial for melanin formation [48]. We next assessed the impact of SIP on ERK, JNK, and p38 for both total and phosphorylated protein expression in B16F10 cells to investigate the potential function of MAPKs in regulating melanogenesis. According to reports, MITF expression is increased by p38 and JNK phosphorylation, which is linked to melanogenesis [67,68,69]. On the other hand, ERK phosphorylation reduces TYR activity and melanin formation in B16F10 cells [70]. According to our findings, SIP only blocked the JNK phosphorylation (Figure 7).

The discovery that the interactivity between MITF and β-catenin is essential to controlling downstream gene expression demonstrated the importance of the β-catenin signaling pathway for melanocyte growth and melanin generation [52,71]. Given that SIP reduced MITF expression in B16F10 melanoma cells, we postulated that Wnt/β-catenin signaling was responsible for this suppression. Interestingly, the findings demonstrate that treatment with SIP substantially and concentration-dependently reduced the intracellular β-catenin expression level, an essential modulating protein of Wnt/β-catenin signaling.

The ubiquitin-proteasome system is an important pathway for the degradation of the intracellular protein. The protein that must be degraded is first tagged with ubiquitin, and the proteasome then specifically recognizes and destroys this ubiquitin-tagged substrate protein. Ubiquitin-controlled protein degradation is physiologically essential for removing erroneous proteins and regulating the cell growth cycle, DNA replication, transcription activation, and gene expression [72]. Previous studies indicate that Ubiquitin-mediated proteolysis is crucial for regulating intracellular β-catenin protein expression levels [51,52]. Therefore, we next investigated if this proteasome degradation contributed to the decreased level of β-catenin caused by SIP. We pre-treated MG132, a proteasome inhibitor, with SIP in order to clarify if β-catenin degradation is mediated by this pathway. Treatment with MG132 nearly entirely recovered SIP reduced β-catenin levels. Accordingly, MG132 evidently also increased the MITF protein level in B16F10 cells in the presence of SIP. In conclusion, these findings point to the involvement of proteasome-mediated degradation of β-catenin in the mechanism by which SIP inhibits Wnt/β-catenin signaling.

When β-catenin accumulates in the cytoplasm of melanocytes, it transfers to the nucleus where it joins with T cell factor (TCF) and lymphoid enhancer factor-1 (LEF-1) to form a complex. The forming complex then acts at the MITF promoter site to increase transcriptional expression, affecting melanin synthesis [29]. Therefore, we assumed that the decreased β-catenin by SIP might have an impact on the release of β-catenin from the cytoplasm to the nucleus. Thus, we measured the protein expression levels of β-catenin both in the cytoplasm and the nucleus of forskolin-induced B16F10 cells. According to the findings, β-catenin content in the nucleus and cytoplasm substantially decreased following SIP treatment, indicating that β-catenin degradation blocks it from migrating from the cytoplasm to the nucleus, interfering with its ability to bind to MITF and suppress melanogenesis.

## 5. Conclusions

In order to investigate the viability of polysaccharides from *S. involucrata* (SIP) as a skin-whitening ingredient, the antioxidant and melanin inhibition effect of SIP and the underlying mechanisms involved were evaluated. SIP was discovered to have a significant *in vitro* radical scavenging activity. The anti-melanogenic effect of SIP was then investigated in forskolin-stimulated B16F10 melanoma cells. The results reveal that SIP remarkably inhibited melanin biosynthesis by downregulating melanogenesis-related proteins and factors. Signaling pathway analysis indicates that the JNK-MAPK signaling pathway and β-catenin signaling involving the proteasome-mediated degradation of β-catenin were also responsible for the anti-melanogenesis effect of SIP. These findings imply that the polysaccharides extracted from *S. involucrata* have strong anti-melanogenic properties and may have applications as a natural functional ingredient for skin-whitening in the cosmetics industry.

## Figures and Tables

**Figure 1 nutrients-14-05044-f001:**
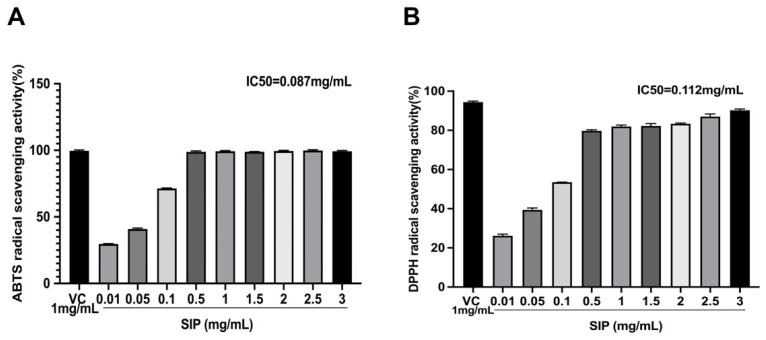
(**A**) ABTS radical scavenging activity of SIP. (**B**) DPPH radical scavenging activity of SIP. Results were presented as the mean ± SD (*n* = 6).

**Figure 2 nutrients-14-05044-f002:**
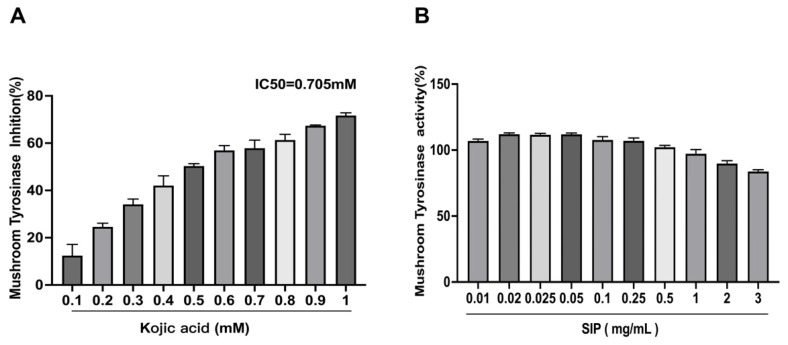
Effect of SIP and positive control kojic acid on mushroom tyrosinase activity. (**A**) Tyrosinase inhibitory effect of kojic acid using L-DOPA as substrate. (**B**) The effect of SIP on tyrosinase activity using L-DOPA as substrate. Results were presented as the mean ± SD (*n* = 6).

**Figure 3 nutrients-14-05044-f003:**
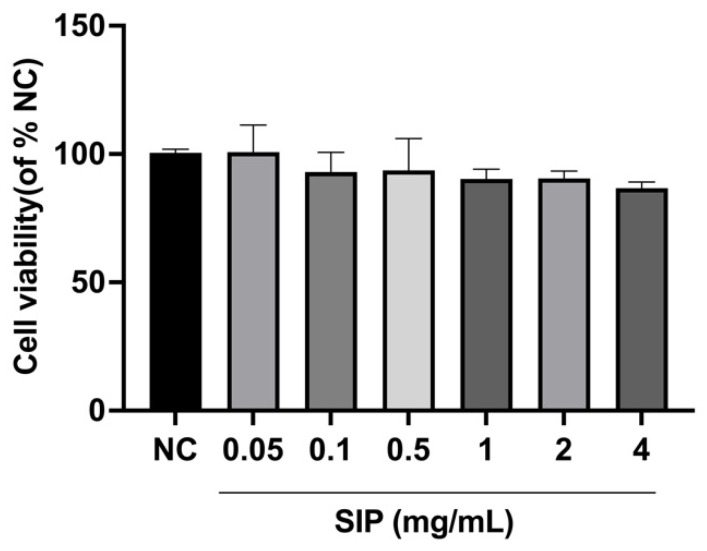
Effect of SIP on B16F10 cell viability. Results were presented as the mean ± SD (*n* = 6).

**Figure 4 nutrients-14-05044-f004:**
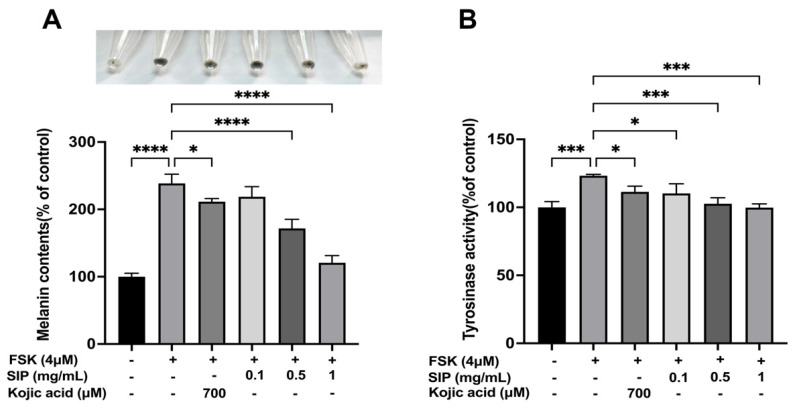
Effects of SIP on the melanin content and tyrosinase activity in FSK-stimulated B16F10 melanoma cells. (**A**) Melanin concentration following SIP treatment for 72 h with the indicated concentration. (**B**) Tyrosinase activity following SIP treatment for 24 h with indicated concentration. Results were presented as the mean ± SD. Statistical analysis was performed with one-way ANOVA, followed by Tukey’s multiple comparison test (*n* = 3; * *p* < 0.05, *** *p* < 0.001, and **** *p* < 0.0001 compared with the forskolin-stimulated group).

**Figure 5 nutrients-14-05044-f005:**
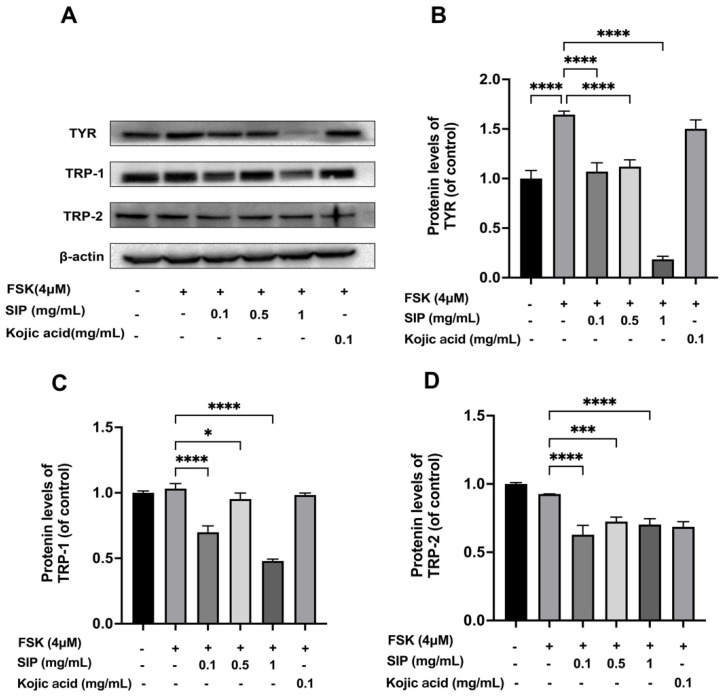
Effects of SIP on melanogenesis-related protein expression levels in FSK-stimulated B16F10 melanoma cells. (**A**) Protein expression levels of TYR, TRP-1, and TRP-2 were determined by Western blot following SIP treatment for 72 h with the indicated concentration. The densitometric analysis of TYR (**B**), TRP-1 (**C**), and TRP-2 (**D**) using Photoshop software2020 (Adobe, San Jose, CA, USA). Results were presented as mean ± SD. Statistical analysis was performed with one-way ANOVA, followed by Tukey’s multiple comparison test (*n* = 3; * *p* < 0.05, *** *p* < 0.001, and **** *p* < 0.0001 compared with the forskolin-stimulated group).

**Figure 6 nutrients-14-05044-f006:**
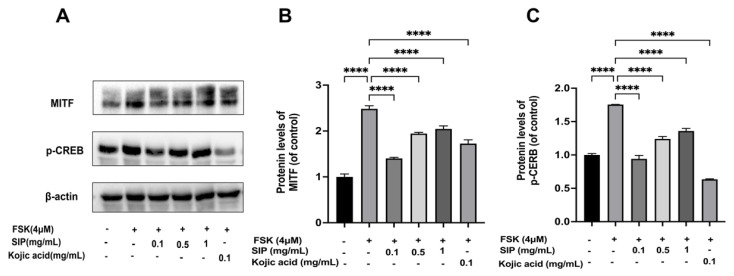
Effects of SIP on MITF and phosphorylated CREB (*p*-CREB) protein expression levels in FSK-stimulated B16F10 melanoma cells. (**A**) MITF and *p*-CREB protein expression levels were detected by Western blot following SIP treatment with the indicated concentration. The densitometric analysis of MITF (**B**), and *p*-CREB (**C**) using the Photoshop software. Results were presented as mean ± SD. Statistical analysis was performed with one-way ANOVA, followed by Tukey’s multiple comparison test (*n* = 3; **** *p* < 0.0001 compared with the forskolin-stimulated group).

**Figure 7 nutrients-14-05044-f007:**
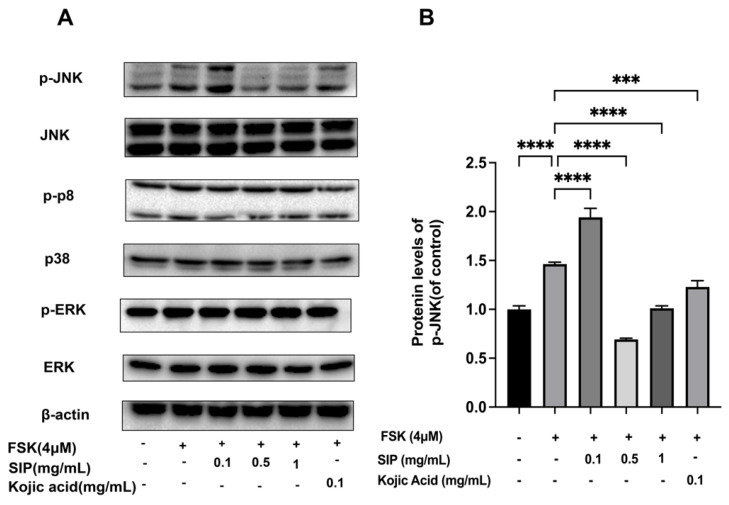
Effects of SIP on MAPK signaling pathway in FSK-stimulated B16F10 melanoma cells. (**A**) Phosphorylation and total Protein expression levels of ERK, p38, and JNK were detected by Western blot following SIP treatment with the indicated concentration. (**B**) The densitometric analysis of *p*-JNK using Photoshop software. Results were presented as mean ± SD. Statistical analysis was performed with one-way ANOVA, followed by Tukey’s multiple comparison test (*n* = 3; *** *p* < 0.001, and **** *p* < 0.0001 compared with the forskolin-stimulated group).

**Figure 8 nutrients-14-05044-f008:**
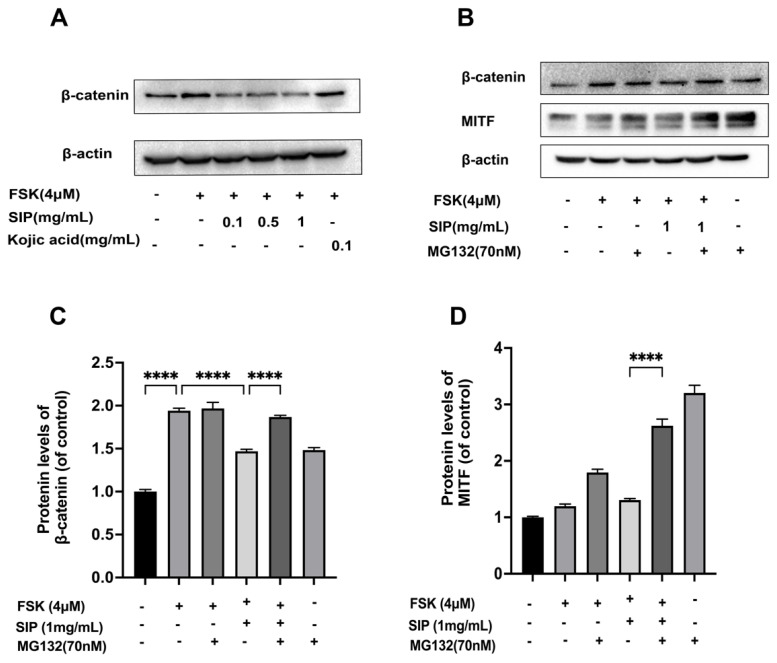
Effects of SIP on the β-catenin signaling pathway in FSK-stimulated B16F10 melanoma cells. (**A**) Protein expression levels of β-catenin were detected by Western blot following SIP treatment for 72 h with the indicated concentration. (**B**) Protein expression levels of β-catenin and MITF were detected by Western blot following preincubation with 70 nM MG132 for 4 h before adding 1 mg/mL SIP and then incubated for another two days. The densitometric analysis of β-catenin (**C**) and MITF (**D**) after MG132 treatment using Photoshop software. Results were presented as mean ± SD. Statistical analysis was performed with one-way ANOVA, followed by Tukey’s multiple comparison test (*n* = 3; **** *p* < 0.0001 compared with the forskolin-stimulated group).

**Figure 9 nutrients-14-05044-f009:**
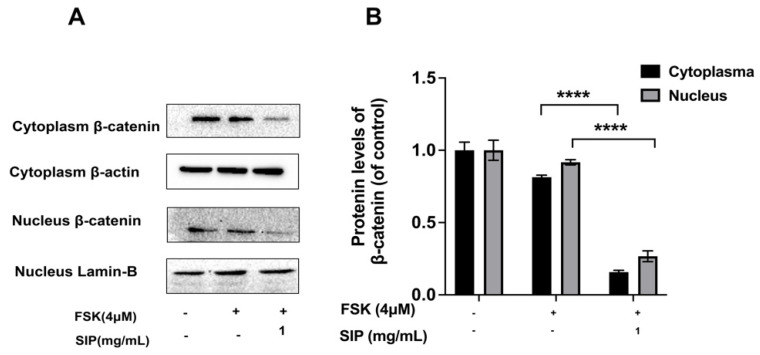
Effects of SIP on β-catenin protein expression levels in the nucleus and cytoplasm of FSK-stimulated B16F10 melanoma cells. (**A**) Protein expression levels of β-catenin in nuclear and cytoplasmic were detected by Western blot following SIP (1 mg/mL) treatment for 48 h. (**B**) The densitometric analysis of β-catenin WB bands using Photoshop software. Results were presented as mean ± SD. Statistical analysis was performed with one-way ANOVA, followed by Tukey’s multiple comparison test (*n* = 3; **** *p* < 0.0001 compared with the forskolin-stimulated group).

## Data Availability

Not applicable.

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
