# Peer review of "Anti-Melanogenesis Effect of Polysaccharide from Saussurea involucrata on Forskolin-Induced Melanogenesis in B16F10 Melanoma Cells"

_nutrients, 2022, doi:10.3390/nu14235044_

Round 1
Reviewer 1 Report
The paper presents information regarding "Anti-Melanogenesis Effect of Polysaccharide from Saussurea involucrata on Forskolin-Induced Melanogenesis in B16F10 melanoma cells".
The manuscript is clearly presented, well documented and the subject is of interest. In order to improve the quality of the manuscript I recommend the authors:
1. The authors should explain the novelty of the work with more details.
2. The grammatical and typo errors should be revised.
3. Both forms are used in the text (μL ; μl). Please change the μl with μL everywhere (e.g. see 2.4.3; 3.4, 3.6, Figure 6, etc.).
4. in vitro please used italic form (Abstract section).
5. rows 147: -20oC please change with -20 oC.
6. rows 193: 100μl please change with 100 μL.
7. Lack of conclusion. The conclusion should be a summarized version of the total manuscript. It is better to summarize what the author has discussed in the manuscript briefly.
I recommend publication of this paper after minor revision.
Author Response
We feel great thanks for your professional and carefully reviewed work on our article, which helped us to improve our paper a lot. As you are concerned, there are several problems that need to be addressed. According to your nice suggestions, we have made extensive corrections to our previous draft, the detailed corrections are listed below.
- The authors should explain the novelty of the work with more details
Response:
We greatly appreciate your constructive suggestion. Indeed, the introduction section of our original draft of this paper did not make the novelty and importance clear. In view of this, we have carefully considered your advice and have strengthened the introduction section to emphasize the novelty. The followings are the novelty of our work in detail:
- As one of the well-known medicinal herbs, Saussurea involucrata has been traditionally demonstrated to have a great number of therapeutic properties including accelerating blood circulation, having neuropharmacological effects, and treating inflammation-related disorders. Flavones, phenylpropanoids, polysaccharides, lignans, and sesquiterpenoids, among other bioactive components found in Saussurea involucrata, have been associated with some specific bioactivities, including antioxidant, anti-UV, and anti-aging. Through an extensive literature review, we found that little research has been reported on the polysaccharides from Saussurea involucrata and its skin pigment-regulating effects. As for this aspect, our work can define its novel property of anti-melanogenesis in addition to its conventional bioactivities.
- Recent cosmetic research has paid attention to creating innovative skin-lighting substances that are safe and efficient melanin inhibitors derived from natural products. Saussurea involucrate was described as an edible Material Medica in the Chinese pharmacopoeia (2020), which prioritized its safe use in cosmetics over other chemicals.
- Our most recent research has proven the powerful anti-inflammatory properties of the polysaccharides from Saussurea involucrata. Therefore, combining its anti-inflammatory benefits with our present whitening effects can provide us a more comprehensive understanding of the functions of Saussurea involucrate in the application of cosmetics.
- To our knowledge, this may be the first time a comprehensive investigation of the anti-melanogenesis activity of polysaccharides derived from Saussurea involucrate has been conducted as well as its underlying mechanism. These findings might provide a reference for the development of comprehensive utilization of S. involucrate particularly the viability of employing SIP as a natural skin-whitening agent in the cosmetics industry.
We added the above details relating to the novelty of our work in the introduction part which were marked in red.
- The grammatical and typo errors should be revised.
Response:
We feel extremely sorry for the grammatical and typo errors we have made in the manuscript. According to your kind advice, we have carefully checked the whole paper and tried our best to revise the grammatical or typographical errors. All the corrections were indicated in the revised manuscript.
- Both forms are used in the text (μL; μl). Please change the μl with μL everywhere (e.g. see 2.4.3; 3.4, 3.6, Figure 6, etc.).
Response:
Thanks for your careful checks. We are really sorry for our careless mistakes. Based on your comments, we have made the corrections to make the unit harmonized within the whole manuscript including all the Figures.
- in vitro please used italic form (Abstract section)
Response:
We sincerely thank you for your careful reading. As suggested, we have corrected the “in vitro” into “in vitro”.
- rows 147: -20℃please change with -20 ℃.
Response:
We feel sorry for our carelessness. We have corrected it as -20 ℃ and we also feel great thanks for your point out.
- rows 193: 100μl please change with 100 μL.
Response:
We have carefully checked the manuscript and corrected the volume unit accordingly. Thanks for your correction.
- Lack of conclusion. The conclusion should be a summarized version of the total manuscript. It is better to summarize what the author has discussed in the manuscript briefly.
Response:
Thank you for reminding us of the importance of the conclusion. As suggested, we have rewritten this part in the revised manuscript. The following is the revised version of the conclusion.
In order to investigate the viability of polysaccharide from Saussurea involucrata (SIP) as a skin-whitening ingredient, the anti-oxidative, melanin inhibition effects of SIP and the mechanisms involved were evaluated in forskolin-stimulated B16F10 melanoma cells. SIP was discovered to have a significant in vitro radical scavenging activity in ABTS/DPPH radical scavenging assay. The anti-melanogenic effects of SIP on forskolin-induced melanogenesis were then investigated in B16F10 melanoma cells, and the results revealed that SIP was effective in concentration-dependently decreasing melanin formation and inhibiting tyrosinase activity. Further research was conducted by measuring the melanogenesis-related proteins and signaling pathway proteins involved in melanogenesis. Melanin accumulation was repressed by SIP via downregulating of the TYR gene family proteins (particularly TYR) and MITF in forskolin-induced melanogenesis. Signaling pathway analysis presented that the JNK-MAPK signaling pathway and β-catenin signaling involving the proteasome-mediated degradation of β-catenin were responsible for the anti-melanogenesis effect of SIP. These findings implied that the polysaccharides extracted from Saussurea involucrata have strong anti-melanogenic properties and may have applications as a natural functional ingredient for skin-whitening in the cosmetics industry.
Reviewer 2 Report
Dear Authors, thank you for this nice work.
Author Response
Thank you for your encouraging comments and kind reminder. According to your suggestion, we corrected the grammatical errors and made an effort to correct the spelling and grammar errors and polish the whole manuscript. We sincerely hope the revised manuscript is understandable to readers.
Reviewer 3 Report
The authors of this manuscript present interesting research on anti-melanogenesis effect of polysaccharide from Saussurea involucrata on Forskolin-induced melanogenesis in B16F10 melanoma cells. Introduction and material and methods are well described. Also, results are presented figures are clear and quite explicable. I believe that the authors to discuss and explain the findings of their work compering their work with other similar works in a well written discussion. The text needs very few revisions. I believed that the presented research work will add further interest to the researchers worldwide.
Abstract
COMMENT:
The abstract describes well the purpose the scope and the fundings of this research work. I suggest once you use latin name of Saussurea involucrata in full use S. involucrata in the rest of the text. Please apply and check authors instructions about this suggestion.
Introduction
Introduction section is well written and, in my opinion, give the appropriate information.
Line 100 Saussurea involucrata
Line 111 Eckonia maxima
Material and Methods
Line 204,220 forskolin (4μM)
Line 232 100 °C
Line 236 4 ËšC
Results
Lines 265, 279,289,323,349,376,399, Delete gap
Discussion
Discussion section is well written.
Conclusions
The conclusion section concludes sufficient the finding of this research work.
References
COMMENT:
Please check reference list according to author’s instruction.
Author Response
We sincerely thank you for your valuable feedback, which we believe helped us to improve the quality of our manuscript. According to your kind suggestions, we have made extensive corrections to our previous draft (all the corrections have been done and marked using the “Track changes “ function in our manuscript ), the detailed corrections are listed below.
- Abstract:
The abstract describes well the purpose the scope and the fundings of this research work. I suggest once you use latin name of Saussurea involucrata in full use S. involucrata in the rest of the text. Please apply and check authors instructions about this suggestion.
Response:
Thank you again for your positive comments and valuable suggestions. Thank you for pointing out the proper use of the Latin name. According to your suggestions, we have checked the instructions for the authors of the journal. We used the full Latin name for the first time in the abstract Saussurea involucrate, and for the rest of the parts, we revised it as S. involucrate for short.
- Introduction
Introduction section is well written and, in my opinion, give the appropriate information.
Line 100 Saussurea involucrata
Line 111 Eckonia maxima
Response:
Line 100 Saussurea involucrata
We deeply appreciate your kind suggestion. According to the comment, we provided more details to describe it as the following:
Our previous study isolated the polysaccharides from Saussurea Involucrate (SIP), and evaluated its anti-inflammatory activities in LPS-stimulated RAW264.7 macrophages. The results showed that SIP possessed a remarkable anti-inflammatory activity by effectively inhibiting the expression of pro-inflammatory cytokines and inflammation-related mediators indicating its potential in functional food, pharmaceutical, and cosmetical industry[1].
Line 111 Eckonia maxima
Sulfated polysaccharides extracted from Eckonia maxima, a kind of brown seaweed, were reported to significantly decreased melanin production and suppressed tyrosinase activity in-MSH-induced B16F10 cells, suggesting their potential use in pharmaceutical and cosmeceutical applications[2].
- Material and Methods
Line 204,220 forskolin (4μM)
Line 232 100 °C
Line 236 4 ËšC
Response:
Thanks for your careful checks. We are sorry for our carelessness. We have already made the corrections in the revised manuscript.
Line 204,220 forskolin (4μM) ---4 µM
Line 232 100 °C---100 ℃
Line 236 4 ˚C---4 ℃
- Results
Lines 265, 279,289,323,349,376,399, Delete gap
Response:
Thanks for your careful checks. We have already deleted the gap in the revised manuscript.
- Discussion
Discussion section is well written.
Response:
Thanks for your encouraging comment.
- Conclusion
The conclusion section concludes sufficient the finding of this research work.
Response:
We really appreciate all your generous comments.
- References
Please check reference list according to author’s instruction.
Response:
Thanks for your kind reminding. We have checked the reference list according to author’s instruction.
References
- Liu, G.; Kamilijiang, M.; Abuduwaili, A.; Zang, D.; Abudukelimu, N.; Liu, G.; Yili, A.; Aisa, H.A. Isolation, structure elucidation, and biological activity of polysaccharides from Saussurea involucrata. International journal of biological macromolecules 2022, 222, 154-166, doi:10.1016/j.ijbiomac.2022.09.137.
- Wang, L.; Jayawardena, T.U.; Yang, H.-W.; Lee, H.-G.; Jeon, Y.-J. The Potential of Sulfated Polysaccharides Isolated from the Brown SeaweedEcklonia maximain Cosmetics: Antioxidant, Anti-melanogenesis, and Photoprotective Activities. Antioxidants 2020, 9, doi:10.3390/antiox9080724.